 

# Real-time observation of signal recognition particle binding to actively translating ribosomes

**Thomas R Noriega**[1,2], **Jin Chen**[3,4], **Peter Walter**[1,2]*, **Joseph D Puglisi**[3]*

[1]Howard Hughes Medical Institute, University of California, San Francisco, San Francisco, United States; [2]Department of Biochemistry and Biophysics, University of California, San Francisco, San Francisco, United States; [3]Department of Structural Biology, Stanford University School of Medicine, Stanford, United States; [4]Department of Applied Physics, Stanford University, Stanford, United States

**Abstract** The signal recognition particle (SRP) directs translating ribosome-nascent chain complexes (RNCs) that display a signal sequence to protein translocation channels in target membranes. All previous work on the initial step of the targeting reaction, when SRP binds to RNCs, used stalled and non-translating RNCs. This meant that an important dimension of the co-translational process remained unstudied. We apply single-molecule fluorescence measurements to observe directly and in real-time *E. coli* SRP binding to actively translating RNCs. We show at physiologically relevant SRP concentrations that SRP-RNC association and dissociation rates depend on nascent chain length and the exposure of a functional signal sequence outside the ribosome. Our results resolve a long-standing question: how can a limited, sub-stoichiometric pool of cellular SRP effectively distinguish RNCs displaying a signal sequence from those that are not? The answer is strikingly simple: as originally proposed, SRP only stably engages translating RNCs exposing a functional signal sequence.

*For correspondence: peter@walterlab.ucsf.edu (PW); puglisi@stanford.edu (JDP)

**Competing interests:** The authors declare that no competing interests exist.

**Reviewing editor**: Ramanujan S Hegde, MRC Laboratory of Molecular Biology, United Kingdom

## Introduction

The signal recognition particle (SRP) in all three kingdoms of life catalyzes the co-translational targeting of membrane and secretory proteins (*Egea et al., 2005*; *Zhang and Shan, 2014*). At the beginning of the targeting reaction, SRP binds to a ribosome-nascent chain complex (RNC). If the RNC displays a signal sequence, RNC-bound SRP binds the SRP receptor at the target membrane (the endoplasmic reticulum membrane in eukaryotes, or the inner membrane in prokaryotes). The membrane-localized RNC is then transferred to the translocon, a protein translocation channel through which the nascent chain passes across, or into, the target membrane.

Whereas mammalian SRP is composed of a 300-nucleotide RNA and 6 protein subunits, the simpler *Escherichia coli* SRP is composed of a 114-nucleotide RNA (4.5S RNA) homologous to a conserved domain of the eukaryotic SRP RNA and a single protein subunit (Ffh), a homolog of the mammalian SRP54 subunit. The *E. coli* SRP, which is used in these studies, can efficiently replace mammalian SRP in *in vitro* targeting reactions, demonstrating that it retains the core targeting functionality (*Bernstein et al., 1993*; *Powers and Walter, 1997*).

Despite wide and careful study, a consistent understanding of the initial step of the targeting reaction, in which SRP binds to translating RNCs, remains elusive. Equilibrium measurements of SRP binding affinities to RNCs stalled with nascent chains up to 35 amino acids in length indicated very tight binding (~1–100 nM binding constants) (*Bornemann et al., 2008*). An estimate based on ribosome profiling of the ~2000 most expressed proteins in *E. coli* indicates that at any given moment

**eLife digest** Genes contain the instructions needed to make proteins from smaller building blocks called amino acids. These instructions are first transcribed to produce molecules of messenger RNA, which are then translated by a ribosome. This 'molecular machine' translates the instructions in the messenger RNA into the sequence of amino acids needed to make the protein.

For some proteins to carry out their role, they need to be delivered to the outside of the cell, or inserted into one of the cell's membranes. As they are being built, these proteins are identified by a so-called 'signal recognition particle', which is often called an SRP for short. The SRP attaches to the new protein when it is still joined to the ribosome, and pulls the protein-ribosome complex to an opening in the target membrane. The new protein chain then enters this opening and either passes through to the other side of the membrane, or ends up embedded within it.

To date, most studies that have investigated this process have involved scientists stalling the building of the new protein to see how SRP interacts with inactivated protein-ribosome complexes. Unfortunately, this means that some of the details of what happens during this process have likely been missed.

Now, Noriega et al. have addressed this problem by developing a method to watch, in real-time, a single active protein-ribosome complex interacting with individual SRPs. This was achieved by attaching fluorescent molecules to SRP and protein-ribosome complexes purified from the bacterium *E. coli*. The distance between the two fluorescent molecules was then tracked over time. This revealed that the SRP typically binds to the protein-ribosome complex after 40–55 amino acids have been built into the protein. At this point, a so-called 'signal sequence' of amino acids has emerged from the complex and can be recognized by the SRP.

Earlier studies had suggested that signal sequences might tell the SRP when to bind, but this had not been demonstrated in experiments using active protein-ribosome complexes. The strategy of using fluorescent molecules to follow single molecules undergoing this process in real-time could now be used by other scientists to re-examine and determine new properties of the protein-ribosome complex in action.

~10% of RNCs have a nascent chain less than 35 amino acids long (*Oh et al., 2011*). Considering that the SRP concentration in *E. coli* is ~400 nM (100-fold less than the ribosomal concentration) (*Jensen and Pedersen, 1994*), such tight binding affinities would result in 75–100% of the SRP to be bound to these RNCs that are not exposing a signal sequence, and the majority of which (~95%) never will (*Bornemann et al., 2008*). SRP binding to these RNCs would thus result in a large unproductive sink on the targeting reaction. Kinetic studies attempted to resolve this issue and concluded that, regardless of nascent chain length, SRP arrives at RNCs very quickly (arrival rates on the order of $10^6$ $M^{-1}$ $sec^{-1}$) and that nascent chain length mostly affects dissociation rates (although different studies have determined a wide range of dissociation rates: ~10–0.01 $s^{-1}$ for RNCs with no nascent chain and ~0.1 to 2 × $10^{-4}$ $s^{-1}$ for RNCs with an exposed signal sequence) (*Holtkamp et al., 2012*; *Noriega et al., 2014*; *Saraogi et al., 2014*). The models that emerged had SRP non-specifically, and quickly arriving to RNCs as soon they begin translating and remaining bound until a nascent chain without a signal sequence becomes long enough to emerge from the ribosomal peptide tunnel and sterically displace SRP. Alternatively, the possibility of an additional factor (such as the co-translational chaperone trigger factor) was proposed to bind the RNC and displace SRP (*Holtkamp et al., 2012*; *Bornemann et al., 2014*). In either model, a large pool of SRP would be unproductively bound for a significant amount of time until displaced.

Prior studies of SRP-RNC binding were performed on RNCs that had been stalled while translating the nascent chain. This approach was technically necessary to create homogenous RNC populations, but lacked the key temporal dimension, provided by active translation by RNCs. Multiple parameters are dynamically changing during active translation: (i) ribosome conformations, which cycle through pre- and post-elongation states, (ii) nascent chain composition, which changes with each new amino acid added, (iii) and length and folding state of the chain outside the ribosomal peptide tunnel. These factors all could affect how SRP interacts with, and subsequently targets, the translating RNCs. Here we developed a single-molecule fluorescence resonance energy transfer (smFRET) assay that allowed

us to observe SRP binding to actively translating RNCs at physiologically relevant SRP concentrations. We show that both association and dissociation rates of SRP binding are sensitive to active RNC translation, with rapid and stable SRP binding to RNCs only upon exposure of a signal sequence outside the ribosomal peptide tunnel.

## Results

### SRP-binding to actively translating RNCs

We used smFRET to detect SRP binding to translating RNCs. To this end, we labeled the 50S ribosome subunit with the FRET donor dye Cy3B at a unique cysteine on ribosomal protein L29 (*Noriega et al., 2014*) and, analogously, SRP with the FRET acceptor dye Cy5 at a unique cysteine in the NG domain of Ffh (*Zhang et al., 2008*; *Shen et al., 2012*). According to all structurally characterized SRP-RNC conformations, these dye positions have inter-dye distances within ~40 and 50 Å (*Halic et al., 2006*; *Schaffitzel et al., 2006*), allowing for detectable FRET between SRP and the ribosome given that the Forster radius of the dyes is ~65 Å (*Uphoff et al., 2010*). To observe SRP binding at relevant concentrations, we performed smFRET experiments using zero-mode waveguides (ZMWs), in which fluorescence measurements are taken from reactions occurring within small metallic apertures (~150 nm in diameter) that are patterned onto a glass substrate (*Chen et al., 2014a*). ZMWs limit background fluorescence from labeled reaction components in solution, which allowed us to measure SRP-RNC binding at 100 nM Cy5-labeled SRP, which is very close to the physiological 400 nM SRP concentration in bacterial cells (*Jensen and Pedersen, 1994*).

We applied the smFRET assay to observe real-time binding of SRP to RNCs actively translating leader peptidase (gene name lepB) mRNA. LepB is a well-characterized in vivo SRP substrate with an N-terminal signal sequence (*de Gier et al., 1996*; *Bornemann et al., 2008*). A 3′-truncated lepB mRNA encoding the first 155 amino acids was immobilized via a biotinylated linker on ZMWs. Pre-initiation complexes ('PICs'; composed of 30S ribosomal subunit, formylated methionine-tRNA$^{fMet}$, and initiation factor 2 in complex with GTP) were then assembled on the mRNA. Finally, we delivered labeled SRP and 50S subunits, as well as a cocktail of unlabeled elongation factors and charged tRNAs (*Johansson et al., 2014*) to the PICs while simultaneously measuring smFRET between SRP and RNCs (*Figure 1A*, top panel).

As shown in *Figure 1A*, we observed a time-resolved image of the translation and SRP recruitment process. Traces of fluorescence as a function of time show substantial and sustained Cy3B fluorescence increase upon delivery of 50S subunits and SRP, indicating translation initiation as the 50S subunit bound stably to the PICs (*green* trace). Following initiation, we observed a stable period (*Figure 1A*, 10–320 s) devoid of apparent SRP-RNC binding events. This time window was followed by a period with extensive FRET events indicating SRP-RNC binding (*Figure 1A*, 320 s and beyond). The E$_{FRET}$ values of the observed smFRET SRP-RNC binding signals were consistent with previous structural and single-molecule characterizations of SRP-RNC complexes (*Figure 1—figure supplement 1*, and 'Materials and methods' '*smFRET assay characterization*').

To quantify SRP-RNC binding events across multiple translating ribosomes, we compared the first SRP arrival times (time between subunit joining and the first SRP-RNC FRET event) to the second to tenth arrival times (time between individual SRP-RNC FRET events). This analysis showed that on average the first SRP binding events were much delayed (~50-fold), compared to subsequent events (*Figure 1B*). The time-dependent progression from a period of no SRP-RNC binding events to one of multiple SRP-RNC binding events is consistent with the original conception of SRP function, which posed that SRP only effectively binds RNCs after the signal sequence on the nascent chain emerged from the ribosomal peptide tunnel (*Walter et al., 1981*).

### SRP-binding to RNCs translating at different rates

To confirm that the results we observed corresponded to SRP binding to actively translating RNCs, we varied the elongation rate. Since elongation rate is directly related to EF-G concentration, RNCs will translate the lepB mRNA more slowly at lower EF-G concentration. We therefore repeated the experiment at two different EF-G concentrations: 750 nM and 250 nM.

We predicted that as RNCs translate more slowly, it would take longer for the signal sequence to become available and hence make observation of multiple SRP binding events during the recorded time course less likely (*Figure 2A*). Indeed, we observed fewer SRP binding events per RNC at the

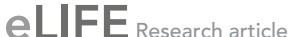

**Figure 1**. SRP-binding to actively translating RNCs. (**A**) Example smFRET trace of Cy5-labeled SRP, Cy3B-labeled 50S subunits, and unlabeled translation mix delivered at time = 0 to PICs pre-assembled on a truncated lepB mRNA (encoding the first 155 amino acids) and immobilized on ZMWs. To reduce non-specific interactions of SRP with the ZMWs, we pre-incubated the ZMWs with BSA, Blocking oligo, and unlabeled SRP, all of which were then
*Figure 1. Continued on next page*

*Figure 1. Continued*

thoroughly washed away (***Figure 1—figure supplement 2***). The top panel shows a schematic representation of the molecular events throughout the trace. The bottom panel shows the fluorescence intensity of the Cy3B (green) and Cy5 (red) signal upon 532 nm excitation. 'AU' indicates 'arbitrary units'. * denotes the initial 50S ribosomal subunit joining. ** denotes photobleaching of the Cy3B dye on the 50S ribosomal subunit. (**B**) Cumulative distributions of SRP first arrival times (blue) and 2nd–10th arrival times (red) to RNCs from the experiment described in (**A**). (n ≥ 141 binding events).

The following source data and figure supplements are available for figure 1:

**Source data 1**. SRP-binding to actively translating RNCs.
**Figure supplement 1**. $E_{FRET}$ validation.
**Figure supplement 2**. ZMW blocking to reduce non-specific SRP interactions.
**Figure supplement 3**. Wide variance in SRP arrival and residence times when RNCs are stalled in translation.

lower EF-G concentration: ~50% of RNCs had more than two SRP binding events in reactions containing 250 nM EF-G, whereas ~80% of RNCs had more than two SRP binding events in reactions containing 750 nM EF-G (***Figure 2B***). Moreover, as expected, the initial SRP-RNC binding events occurred later at lower EF-G concentrations: the half-time of first arrival was ~350 s in 250 nM EF-G reactions and ~280 s in 750 nM EF-G reactions (***Figure 2C***). SRP arrival events that followed the initial binding event were similarly slower at the lower EF-G concentrations: half-time of arrival was ~12 s in 250 nM EF-G reactions and ~5 s in 750 nM EF-G reactions (***Figure 2D***). These latter results are explained because it takes slowly elongating RNCs longer to translate nascent chains in which the signal sequence is optimally exposed, allowing SRP to bind most effectively (***Noriega et al., 2014***).

From previous calibrations of translation rates under single-molecule conditions (~5–10 s per amino acid, at the elongation factor concentrations used) (***Uemura et al., 2010***), we estimate that the first SRP arrival times occurred after approximately 35–45 amino acids were polymerized. This nascent chain length would correspond to the partial emergence of a signal sequence from the ribosome (***Houben et al., 2005***; ***Bornemann et al., 2008***).

## SRP binding to translation-calibrated RNCs

To measure nascent chain length directly, we calibrated the extent of translation by observing labeled tRNA transit events during translation (***Figure 3***). To this end, we used an engineered 3'-end truncated lepB mRNA encoding 95 amino acids. The mRNA contained a single phenylalanine codon followed by three clusters of three sequential phenylalanine codons as chain length markers at positions 5, 25–27, 55–57, and 85–87 (***Figure 3—figure supplement 1***, cWT for 'calibration WT'). After replacing unlabeled phenylalanine tRNA (tRNA^Phe) in the translation mix with Cy3.5-labeled tRNA^Phe and monitoring Cy3.5 fluorescence, we observed pulses in fluorescence intensity when phenylalanine was incorporated into the nascent chain (***Chen et al., 2014b***; ***Tsai et al., 2014***). The duration of the pulses represent the transit of tRNA^Phe through two rounds of peptide elongation and departure, whereas interpulse durations represent the translation time between the phenylalanine codons.

This experimental set-up consistently yielded clear tRNA^Phe pulses superimposed on SRP-RNC FRET binding events (***Figure 3A***). The tRNA^Phe pulses behaved as expected: (i) The majority of RNC traces that showed SRP binding displayed three or four tRNA^Phe pulses (***Figure 3—figure supplement 2***, upper left panel); (ii) the second, third, and fourth tRNA pulses, during which clusters of three phenylalanines were incorporated, showed the same average lifetimes (~30 s) and lasted three-times longer than the average lifetime of the first pulse (~10 s), in which only a single phenylalanine was incorporated (***Figure 3—figure supplement 2***, upper right panel); and (iii) the inter-pulse times were proportional to the number of codons translated between them (***Figure 3—figure supplement 2***, lower panels). These results indicate that the RNCs under our assay conditions are actively translating and that we can accurately calibrate the reaction to map SRP binding events on nascent chain length.

Using this assay, we next determined the distribution of initial SRP-RNC binding events. We observed that the majority of first binding events occurred when RNCs had translated between 40 and 55 amino acids (68% of all events) (***Figure 3C***, ***Figure 3—figure supplement 3***), corresponding to a nascent chain length at which the signal sequence emerges from the ribosomal peptide tunnel

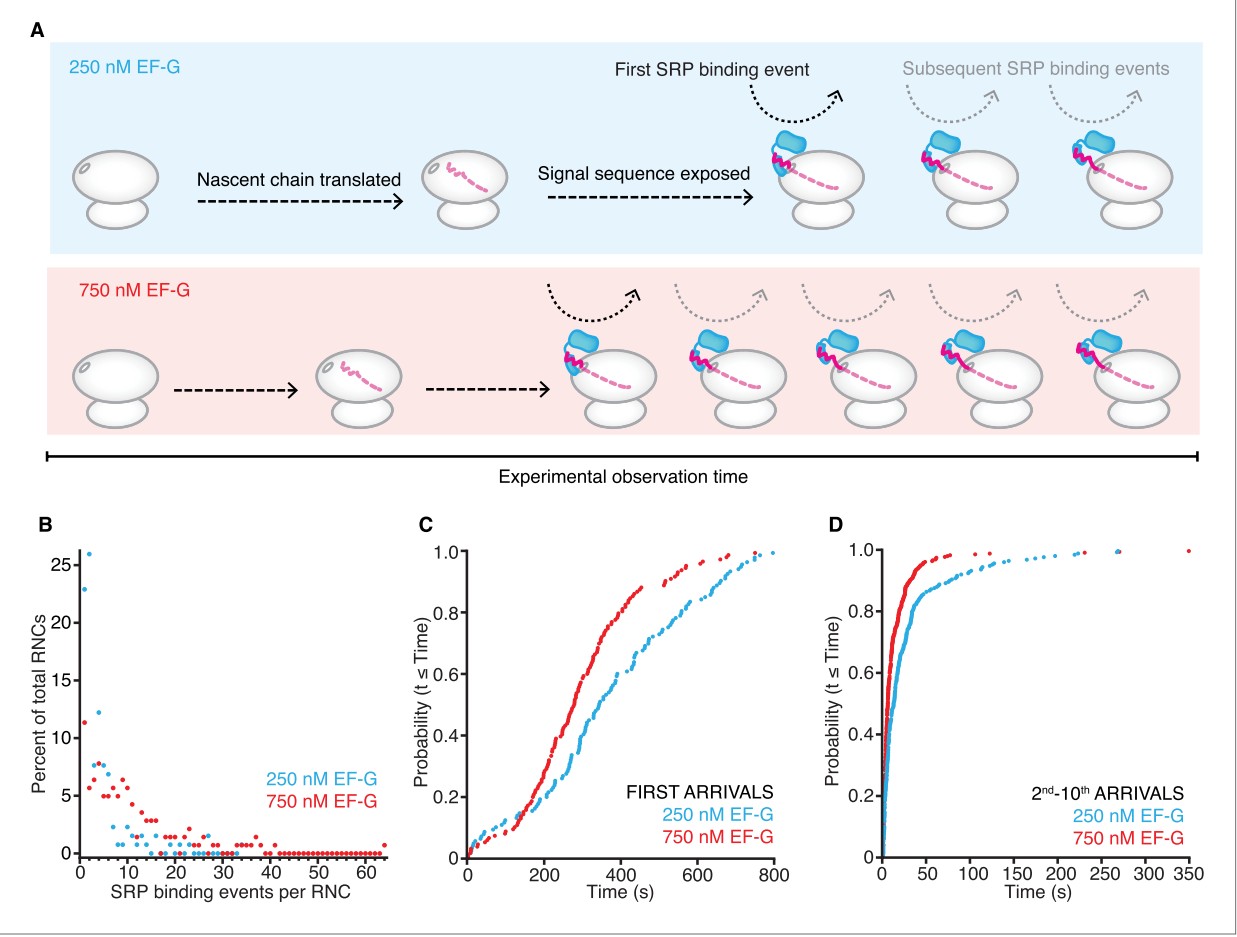

**Figure 2**. SRP-binding to RNCs translating at different rates. (**A**) Schematic representation of the effect on the number of SRP-RNC binding events and their first and subsequent arrival times when performing the experiment described in **Figure 1** at 250 nM (blue) and 750 nM (red) EF-G concentrations. (**B**) Comparison of SRP-binding events per RNC distributions for the experiment described in **A**. Colors as in (**A**) (n ≥ 621 binding events) (**C**–**D**) Cumulative distributions of SRP first arrival times (**C**) and 2nd–10th arrival times (**D**) at 750 nM (red) and 250 nM (blue) (n ≥ 131 binding events).

The following source data is available for figure 2:

**Source data 1**. SRP-binding to RNCs translating at different rates.

(**Houben et al., 2005**; **Bornemann et al., 2008**). Only 9% of observed first arrivals occurred before the nascent chain was 40 amino acids long. To confirm the dependence of SRP-RNC binding on the presence of a functional signal sequence, we measured SRP binding in a reaction translating lepB with a mutated signal sequence, previously shown to be inactive (**Houben et al., 2005**) (**Figure 3—figure supplement 1**, cMT for 'calibration mutant'). SRP binding events were virtually absent upon translation of the cMT lepB mRNA (**Figure 3C,E**).

We also observed that, past a 50 amino acid nascent chain length, SRP-RNC binding events occurred in quick succession (at a rate of 1 per ~1–2 codons translated *after* 50 amino acid chain length, as opposed to 1 per ~50 codons *before* 50 amino acid chain length) (**Figure 3C**, **Figure 3—figure supplement 3**), consistent with the ~50-fold increase in SRP association rates upon exposure of a functional signal sequence shown above in **Figure 1B**. We also observed that SRP-RNC residence times were dependent on translation. Residence times were longest when RNCs were translating nascent chains in the 50–60 amino acid range (**Figure 3C**, red squares and dashed line; average lifetimes of 74.4 ± 5.0 s). Shorter or longer nascent chain lengths resulted in ~two-fold briefer SRP-RNC residence times (**Figure 3C**, red squares and dashed line; average lifetimes of 34.7 ± 4.1 and 37.8 ± 3.5 s for chains in the 40–50 and 60–70 amino acid range, respectively). These results are consistent with

**Figure 3**. SRP binding to translation-calibrated RNCs. (**A**) Representative smFRET trace of Cy5-labeled SRP, Cy3B-labeled 50S subunits, Cy3.5-labeled F-tRNA and unlabeled translation mix delivered at time = 0 to PICs pre-assembled on a lepB cWT mRNA (encoding the first 95 amino acids) and immobilized on ZMWs (see text and ***Figure 3—figure supplement 1***). Fluorescence intensity of the Cy3B (green), Cy3.5 (orange), and Cy5 (red) signal under 532 nm excitation are shown. 'AU' indicates 'arbitrary units'. * denotes the initial 50S ribosomal subunit joining. ** denotes photobleaching of the Cy3B dye on the 50S ribosomal subunit. (**B**) Schematic showing when, relative to the x-axis shared by panels **C**–**E**, the signal

*Figure 3. Continued on next page*

*Figure 3. Continued*

sequence is exposed from RNCs. (**C**) Histogram showing how many amino acids have been polymerized when SRP first arrives to RNCs actively translating the lepB cWT mRNA (blue) or cMT mRNA (orange). Y-axis shows both total events, and percent of total for RNCs translating cWT mRNA. Note x-axis is shared by (**C–E**). (**D**) Scatter plot of SRP-RNC binding residence times relative to the number of amino acids polymerized when the event starts (black dots), and average lifetimes of the residence times between the tick-marks (red squares and dashed red line, with associated error bars that are too small to be seen). mRNA translated is lepB cWT. Note that for clarity the y-axis is split at 100 s, as indicated by the dashed grey line. Only traces in which four tRNA pulses were detected were included in the analysis in this panel and panel (**E**). (**E**) Histogram showing how many RNCs are occupied by SRP relative to the number of amino acids translated when RNCs are actively translating a lepB cWT mRNA (blue) or cMT mRNA (orange—with so few events that, at this y-axis scale, they are not visible). Y-axis shows both total events, and percent of total for RNCs translating cWT mRNA.

The following source data and figure supplements are available for figure 3:

**Source data 1**. SRP binding to translation-calibrated RNCs.

**Figure supplement 1**. Translation-calibration lepB mRNA constructs.

**Figure supplement 2**. tRNA pulse characterization.

**Figure supplement 3**. Translation-calibrated SRP arrivals to RNCs.

**Figure supplement 4**. SRP binding before and after each tRNA pulse.

SRP binding most effectively (with the longest residence time) to nascent chains of an optimal length (***Noriega et al., 2014***).

As nascent chains grew from 45 to 60 amino acids, we observed a rapid increase in the RNC occupancy by SRP (where occupancy refers to the number of RNCs that are bound by SRP, with RNCs grouped by amino acids translated) (***Figure 3D*** and ***Figure 3—figure supplement 4***), which depended on a functional signal sequence. When the signal sequence was mutated, occupancy was lost.

## Discussion

In this work, we applied the power of single-molecule approaches to observe directly the dynamics of SRP-RNC interaction on actively translating mRNAs. We show that under close to physiological conditions, SRP-RNC interactions change as the nascent chain grows and a signal sequence becomes exposed. Traditional analyses of binding rates, previously deduced from binding reactions using static, stalled RNCs, demonstrated that SRP-RNC binding kinetics are sensitive to nascent chain lengths but yielded conflicting results when describing SRP occupancy (***Siegel and Walter, 1988***; ***Bornemann et al., 2008***; ***Holtkamp et al., 2012***; ***Noriega et al., 2014***). The results presented here establish a new experimental paradigm in which the association of numerous other factors that interact co-translationally with RNCs, such as chaperones and nascent chain modifying enzymes, can be characterized dynamically.

Our results suggest that SRP does not engage stably with translating RNCs that do not expose a functional signal sequence. These results are consistent with past biochemical and structural work showing that SRP can directly bind to an exposed signal sequence (***Zopf et al., 1990***; ***Keenan et al., 1998***; ***Janda et al., 2010***). However, the conclusions contrast with those of previous work on stalled ribosomes, which suggested that SRP can bind prominently to RNCs with nascent chains as short as 20 amino acids containing a signal sequence still occluded within the ribosome (***Bornemann et al., 2008***; ***Holtkamp et al., 2012***). We determined that the kinetic parameter most responsible for the observed signal sequence discrimination is a highly variable SRP-RNC association rate. When no signal sequence is exposed on RNCs ($>3.5 \times 10^4$ $M^{-1}$ $s^{-1}$, for the experiments performed at 750 nM EF-G), we observed only negligible SRP occupancy. After translation advances far enough to have a signal sequence exposed on an RNC, the SRP association rates become at least 50-fold faster ($\sim 1.8 \times 10^6$ $M^{-1}$ $s^{-1}$, for the experiments performed at 750 nM EF-G).

This contrasts with previous work suggesting that stable SRP-RNC complex association rates are insensitive to nascent chain length (***Holtkamp et al., 2012***; ***Saraogi et al., 2014***). Our system is based

on a FRET signal between dyes on the ribosome and SRP, limiting detection to interactions in which the inter-dye distance is less than ~95 Å (which would yield an expected $E_{FRET}$ of ~0.1). Additionally, our experiments have a 100 ms temporal resolution, and no binding would be detected, even if there was a FRET signal, if binding events had residence times much shorter than 100 ms. Given that the known SRP-RNC structures predict inter-dye distances of less than 50 Å (*Halic et al., 2006*; *Schaffitzel et al., 2006*), and the shortest reported average residence time for an SRP-RNC binding event is ~70–100 ms (*Holtkamp et al., 2012*), we are confident that we would detect binding events that are conformationally similar to the known SRP-RNC complexes. However, it is possible that there are intermediates in the binding reaction (including shot-lived, unproductive diffusional collision events) that we would not detect due to distance and temporal resolution limitations. Such intermediates could be encounter complexes previously observed to form with ~$10^6$ $M^{-1}$ $s^{-1}$ association rates, regardless of nascent chain length (*Holtkamp et al., 2012*; *Saraogi et al., 2014*). However, if such intermediates formed in our reaction, they would need to be conformationally distinct from known SRP-RNC structures, or they would have been detected in this work.

An alternative explanation for the discrepancies of our results to the previous work is that they may arise from differences between actively translating RNCs and stalled RNCs. Our assays show that when SRP binding events were compared among stalled RNCs we observed an enormous range in the variance of their arrival and residence times (2.4–40.9 s and 0.4–58.7 s median arrival and residence times, respectively; *Figure 1—figure supplement 3*, and 'Materials and methods'). This variability indicates that individual stalled RNCs may exist in numerous different conformational states, many of which are likely to be inactive and perhaps off-pathway, despite each displaying the same length nascent chain. These data argue that quantitative results obtained with purified and stalled RNCs may be less physiologically relevant than results obtained with similarly purified but actively translating RNCs, which, by the very nature of the assay, represent functional states.

Studies in yeast and mammalian cell observed some preference of SRP for RNCs carrying a signal sequence before it was exposed from the ribosome (*Berndt et al., 2009*; *Mariappan et al., 2010*). These studies used stalled RNCs in crude cell extracts, suggesting that perhaps factors absent in our assays and/or differences between prokaryotic and eukaryotic systems might influence early SRP-RNC interactions and affect signal sequence discrimination. The tools presented here are an important step towards quantitatively testing such a possibility with actively translating ribosomes. More generally, the methods presented here promise to be useful in studies of other RNC-associating factors, including nascent-chain modifying enzymes and co-translational chaperones.

Our results resolve a long-standing question in the field: how can a limited, sub-stoichiometric pool of cellular SRP effectively distinguish RNCs that display a signal sequence from those that do not? The answer appears strikingly simple: as originally proposed (*Walter et al., 1981*) and here confirmed using dynamic single-molecule measurements at physiologically relevant concentrations, SRP only engages translating RNCs that expose a functional signal sequence.

## Materials and methods

### Reagent cloning, expression and purification

All proteins used in this study were derived from *E. coli* strain MC4100 and expressed in *E. coli*. The Ffh and L29 expression constructs and purification protocols have been described previously (*Noriega et al., 2014*). The Ffh(E153C) single-cysteine mutant was engineered using the QuikChange mutagenesis kit (Agilent, Santa Clara, CA). The DNA templates for the calibration lepB mRNAs (cWT and cMT) were ordered as GeneART oligos (Invitrogen) and cloned into pCR-Blunt II-TOPO vector according to the Zero Blunt TOPO kit protocol (Invitrogen, Carlsbad, CA). The mRNA transcripts were in vitro transcribed and prepared for single molecule immobilization as described before (*Noriega et al., 2014*). Blocking oligos (5′-CGTTTACACGTGGGGTCCCAAGCACGCGGCTACTAGATCACGGCTCAGCT-3′, and its reverse complement) were annealed in 50 mM TrisAcetate (pH 7.5 at 25°C) and 100 mM KCl by heating to 95°C for 1 min and then cooling down to 25°C on the bench. All chemicals, unless otherwise stated, where purchased from Sigma (St. Louis, MO).

### Reagent labeling with fluorescent probes

Ffh(E153C) and L29(Q38C) single cysteine variants were labeled with Cy5 and Cy3B, respectively as described previously (*Noriega et al., 2014*). Labeling efficiency was typically >90% for both proteins.

Labeled Ffh and L29 were reconstituted into SRP and RNCs as previously described (*Noriega et al., 2014*).

## smFRET assay characterization

We tested binding of SRP to RNCs stalled on a 3'-truncated mRNA encoding the first 75 amino acids of lepB. This RNC construct has been shown to bind SRP robustly under single-molecule conditions (*Noriega et al., 2014*). When the dye-labeled SRP and RNCs were incubated together to allow SRP-RNC complex formation, smFRET was observed with 0.2–0.32 efficiency ($E_{FRET}$) (*Figure 1—figure supplement 1*, left panel). To calibrate these $E_{FRET}$ values, we performed the same experiment using total internal reflection microscopy (TIRFM), which does not quench the Cy5 signal as the aluminum walls of the apertures in the ZMW set-up do (*Chen et al., 2014a*). The calibration revealed that the observed signal corresponded to a corrected value of 0.33–0.5 $E_{FRET}$ (*Figure 1—figure supplement 1*, right panel). This is lower than the expected $E_{FRET}$ of ~0.85 predicted from molecular modeling onto cryo-EM structures of SRP bound to RNCs (*Halic et al., 2006*; *Schaffitzel et al., 2006*). However, these are relatively low resolution structures (9.6–16 Å), and our previous work has shown that SRP can take on a variety of conformations on the ribosome (*Noriega et al., 2014*), consistent with the observed bimodality in the $E_{FRET}$ distributions (*Figure 1—figure supplement 1*). The $E_{FRET}$ signals are too close to each other to be clearly resolved, so we did not pursue conformational distinctions further in this study. We also compared individual RNCs based on the arrival and residence times of SRP binding events. To do this we immobilized PICs on a truncated mRNA encoding the first 55 amino acids of lepB. We then delivered our single molecule translation mix (as described in the section below) and allowed RNCs to translate for 20 min at room temperature, which based on our active translation experiments, is enough time to ensure full translation of the mRNA. We then delivered 100 nM of Cy5 labeled SRP in to the stalled RNCs and visualized SRP-RNC binding events. This assay allowed us to determine that stalled RNCs, despite being homogeneously stalled with a 55 amino acid nascent chain, show as much as 1–3 orders of magnitude differences in the median and variance of SRP arrival and residence times (*Figure 1—figure supplement 3*). This broad variance among individual RNCs suggests that simply stalling RNCs induces a variety of different RNC conformations that affect SRP-binding, many of which may be physiologically irrelevant because the RNCs are not actively translating.

## Single-molecule SRP delivery experiments and analysis

The experiments with stalled RNCs were performed as described before (*Noriega et al., 2014*). The ZMW delivery experiments with translating RNCs were performed as previously described (*Chen et al., 2014b*; *Johansson et al., 2014*; *Tsai et al., 2014*) with the following modifications: the Tris-based polymix translation buffer was supplemented with 5 mg/ml of Ultrapure BSA (Ambion, Carlsbad, CA) and 10 µM Blocking oligo. Other standard blockers such as PEG, poly-L-lysine, aprotinin, kappa-, and beta-Casein had no effect on non-specific SRP-ZMW interactions. An additional ZMW chip preparatory step was also added after immobilization of the PICs: 20 µl of wash solution supplemented with 400 nM of unlabeled SRP was added to the ZMW chip for 3 min, removed and then rinsed and washed with 20 µl of wash solution without unlabeled SRP. For all ZMW experiments, the concentration of unlabeled charged tRNAs, EF-Tu, and GTP ternary complexes was 2.45 µM. For experiments with labeled tRNA$^{Phe}$, the concentration of tRNA$^{Phe}$ ternary complexes was 200 nM. The concentration of EF-G was 750 nM unless otherwise stated. None of the SRP-RNC binding kinetics curves presented in the figures (except for *Figure 3C*) fit single or double exponential curves. This is expected given that active translation is a complex multi-step process. We did not attempt to fit these curves to a theoretical equation. Instead, when rate estimates were necessary, we compared values at which the curves reached 50% of the measured effects. The data were analyzed using custom Matlab scripts, similar to those described previously (*Chen et al., 2014b*; *Johansson et al., 2014*; *Noriega et al., 2014*; *Tsai et al., 2014*), and made available at: https://github.com/trnoriega/Matlab-Single-Molecule.

## Acknowledgments

We thank Juliet Girard, Margaret Elvekrog, Alexey Petrov, and the Walter and Puglisi labs for helpful discussion and comments. The work presented here was supported by US National Institutes of Health grants GM51266, GM099687 (JDP) and GM032384 (PW) as well as National Institute of General Medicine Initiative for Maximizing Student Development and National Science Foundation Graduate

Research Fellowships (TRN), and Stanford Interdisciplinary Graduate Fellowship (JC). PW is an Investigator of the Howard Hughes Medical Institute.

## Additional information

### Funding

| Funder | Grant reference number | Author |
| --- | --- | --- |
| National Institutes of Health | GM51266 | Jin Chen, Joseph D Puglisi |
| Howard Hughes Medical Institute | Investigator | Peter Walter |
| National Science Foundation | Graduate Research Fellowship | Thomas R Noriega |
| Stanford University | Interdisciplinary Graduate Fellowship | Jin Chen |
| National Institutes of Health | GM099687 | Jin Chen, Joseph D Puglisi |
| National Institutes of Health | GM032384 | Thomas R Noriega, Peter Walter |
| National Institute of General Medical Sciences | Initiative for Maximizing Student Development Fellowship | Thomas R Noriega |

The funders had no role in study design, data collection and interpretation, or the decision to submit the work for publication.

### Author contributions

TRN, Conception and design, Acquisition of data, Analysis and interpretation of data, Drafting or revising the article; JC, Acquisition of data, Analysis and interpretation of data, Drafting or revising the article; PW, JDP, Conception and design, Analysis and interpretation of data, Drafting or revising the article

### Author ORCIDs

Jin Chen, http://orcid.org/0000-0002-6634-4397

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
