## [Decision Letter]

Thank you for sending your work entitled “Real-time observation of signal recognition particle binding to actively translating ribosomes” for consideration at *eLife*. Your article has been favorably evaluated by Randy Schekman (Senior editor) and 3 reviewers, one of whom is a member of our Board of Reviewing Editors.

The Reviewing editor and the other reviewers discussed their comments before we reached this decision, and the Reviewing editor has assembled the following comments to help you prepare a revised submission.

The reviewers all agreed that this was a technically excellent study, that the problem is interesting, and that the results are of interest to the readership of *eLife*. Quoting from one referee: “For many years stalled ribosome-nascent chain complexes (RNCs) have been used to obtain “snapshots” of various co-translational phenomena. Because translation is a highly dynamic process, however, it has never been clear that interactions between static RNCs and targeting factors, chaperones, and other nascent chain binding proteins are physiologically significant. In the present study Noriega et al. overcome this problem by using an elegant and very innovative single molecule FRET approach to examine the binding of E. coli SRP to actively translating ribosomes. They show convincingly that SRP binds to RNCs only when a signal peptide is exposed, and thereby provide the most definitive evidence to date in support of a model that was proposed over 30 years ago. They describe an important and very impressive technical advance that will doubtless be of very broad interest.”

After discussion of the main points raised by the referees, two in particular were highlighted as substantial:

1) The presentation of the context for this work and depth of discussion of the results should be improved. The referees each took issue about different aspects of the text and have made suggestions for improvement. [Editors’ note: the most substantive suggestions are shown in the accompanying Author response.]

2) The second issue relates to how stalled versus translating RNCs were experimentally compared. Quoting from one of the referees: “The authors attribute the differences between their results and those with stalled RNCs to the fact the latter are not translating. There may be other reasons in addition to or instead of this. For example, stalling translation with a hydrophobic TMD exposed to bulk aqueous solvent in a PURE translation system may well result in aggregation, sticking to whatever is available, and other non-physiologic consequences. This seems far more likely to cause the observed heterogeneity and artifacts than the fact that ribosomes are stalled. I therefore think the comparison between the translating ribosomes with smFRET and stalled RNCs prepared by different methods and deposited into the ZMWs afterwards is flawed. The appropriate comparison would be to use truncated mRNA for 55-mer that is translated in the ZMW (exactly as the real-time experiment), then add the SRP after the translation. This would more directly compare stalled versus translating.” Given that a major conclusion put forth relates to this comparison, the proposed experiment would seem worthwhile and well within the reagents and tools available to the authors.

---

## [Author Response]

*1) The presentation of the context for this work and depth of discussion of the results should be improved. The referees each took issue about different aspects of the text and have made suggestions for improvement*.

We have made significant changes to the manuscript based on the reviewers’ suggestions:

*Some of the arguments were exaggerated or mischaracterized unnecessarily. For example, the potential problem of ribosomes serving as a sink for SRP would presumably only apply to ribosomes that have just begun translation (i.e., short nascent chains less than 35 residues), not all cellular ribosomes as implied. Thus, the appropriate comparison is between SRP abundance and the abundance of short RNCs in the cell. The latter information can be inferred from existing ribosome profiling data, and is presumably a minority. This comparison may well be plausible physiologically*.

We agree that our tone was unnecessarily exaggerated, and have modified the introduction to make it clear that we do not think that every RNC in the cell can act as a sink for SRP binding, only those with short nascent chains that aren’t exposing a signal sequence.

We have also incorporated the reviewer’s helpful suggestion and estimated the amount of RNCs that might serve as a sink to the targeting reaction due to their short nascent chain (∼10% based on conservative analysis of Figure 1 in the [18] Cell ribosome profiling paper, which is now cited.)

*The results with slowed translation were somewhat puzzling to me*. *I of course understand why the first arrival of SRP is delayed, but why are subsequent arrivals delayed? After the first, the signal is already outside the tunnel, so why should release and re-binding be affected by translation rate?*

This observation is explained by our previous publication ([17] JBC) showing that even after the signal sequence is exposed the arrival rates to RNCs are different, probably due to there being an ‘ideal’ nascent chain length at which both SRP-RNC arrival and residence times are optimal. We have clarified this in the manuscript.

*The authors observed 9% first arrivals before 40 residues are synthesized. Why are these events lost when the signal peptide is mutated? Shouldn't this remain the same*, *with the only difference observed at lengths after the signal emerges from the ribosome?*

We agree that under ideal circumstances the only difference should be observed at lengths after the signal sequence emerges from the ribosome. The discrepancy is due to some of the mentioned arrivals being an artifact of inaccurate assignment of tRNA events in some of the traces. While our controls show that assignments are mostly accurate there is always the possibility that we miss one of the F-tRNA events. This results in some calibrations yielding too- early estimates of SRP binding. In the case of the arrival analysis we decided to include traces with 2, 3 and 4 tRNA events because, even though it leads to a small loss of accuracy, it allowed us to get a large number of 1st, 2nd, and 3rd events to analyze and compare. To verify that inaccuracy in tRNA assignment is indeed responsible for the discrepancy, we analyzed first event arrivals in traces with 4 tRNA events (and thus less likely to have a missed tRNA event) and found that only 2% happen before 40 amino acids were translated.

*I feel the authors should be more cautious about making sweeping declarations such as 'stalled RNCs are less physiologically relevant than results obtained with actively translating RNCs.” The details of the experiments matter, and it is by no means obvious whether stalled RNCs in complete cytosol is less or more physiologic than a purified system with multiple mutated/labeled components carried out on a glass surface requiring blocking with BSA, etc. A more tempered discussion of the different approaches would be better*.

We have tempered our discussion of the benefits of using actively translating ribosomes as opposed to stalled ribosomes.

*My most significant concern is that the authors tell their story in a rather awkward fashion by suggesting that they have solved a “long-standing paradox”. On the contrary, their main accomplishment has been to prove the prevailing, but inadequately tested, view. It has always seemed logical that SRP binds tightly to RNCs after a signal peptide is exposed because it has a high affinity for signal peptides, and a good deal of biochemical and structural evidence (none of which is cited) is consistent with this idea. A couple of recent studies that have attempted to determine SRP-RNC binding constants have certainly generated some controversy and perhaps illustrate the pitfalls of using static RNCs, but I wouldn't call this a “paradox”. Because there has been some inconsistency in the literature a more definitive approach has been needed, and that's exactly what the authors have delivered. I strongly recommend that they modify the Abstract, Introduction and Discussion to emphasize that they have proven a long-standing model and, perhaps even more importantly, developed a new method of great general utility, rather than focus on a few puzzling recent results*.

We agree with the reviewer that it has always seemed logical that SRP should bind more tightly to RNCs after a signal sequence is exposed. We have now included references for the work showing biochemically and structurally that SRP interacts directly with the signal sequence. We also agree that we should highlight our method’s general utility. We have modified the discussion to reflect these excellent suggestions. The paradox, (or ‘question’ as we now call it) addressed by our paper is: “how can a limited, sub-stoichiometric pool of cellular SRP effectively distinguish RNCs displaying a signal sequence from those that are not?” We think that framing the paper in this way is appropriate since we cite 5 high-profile papers spanning the last 6 years that argue that SRP is somehow distinguishing RNCs while still binding to those that aren’t actually exposing a signal sequence. We are comforted by the reviewer’s statement that our model is the prevailing one in the field, despite other published alternatives.

*The framework used to describe and discuss binding in this manuscript, as well as to interpret and discuss the results of the binding experiments, is overly simplistic. Generally speaking, binding typically proceeds along a pathway involving the collision of the binding partners, the formation of a non-specific encounter complex, the formation of one or more binding intermediates, and the formation of the final complex. This is an important consideration because the authors state that “the kinetic parameter most responsible for signal sequence discrimination is a highly variable SRP- RNC association rate”, but there is no attempt to rationalize this observation within the context of a typical binding pathway. At minimum, one would expect SRP to collide and form non-specific encounter complexes with all RNCs, regardless of whether they contain an exposed signal sequence or not. It is only after the formation of such a non-specific encounter complex that, presumably through the formation of subsequent binding intermediates or lack thereof, SRP would be able to discriminate between RNCs that are carrying an exposed signal sequence and those that are not. In the case of RNCs that do not contain an exposed signal sequence, one would expect that SRP presumably fails to proceed along the pathway to formation of the final complex and instead dissociates rapidly from one or more binding intermediates along the pathway, intermediates whose average lifetimes are too short relative to the 100 msec/frame time resolution of the smFRET measurements to be easily observed. I suspect, however, that if the time resolution of the smFRET measurements were higher, the authors would be able to observe such binding intermediates. Indeed, the observation that the residence times of SRP on RNCs containing exposed signal sequences between 40-50 amino acids and 60-70 amino acids are two-fold shorter than that on RNCs containing exposed signal sequences between 50-60 amino acids imply that binding intermediates with average lifetimes shorter than that of the final complex are indeed populated. Similarly, instances of SRP binding to RNCs containing nascent chains that do not contain a signal sequence, which I infer can be observed, albeit rarely, from the statement that “SRP binding events were virtually absent upon translation of the cMT lepB mRNA”, also imply that such binding intermediates do form. Discussion of the results within this framework will likely have implications for previous studies in which variability in the rate of SRP dissociation from the RNC was found to be the kinetic parameter most responsible for signal sequence discrimination as well as for more recent studies that suggest important roles for more sophisticated mechanisms (e.g., induced fit, kinetic proofreading, editing, etc.), many of which necessarily involve the formation of one or more binding intermediates. In any case, I think the authors should use a slightly more sophisticated framework to describe the binding pathway and use this framework to interpret and discuss their results*.

The reviewer makes an excellent and thoughtful point. Our discussion glossed over SRP first forming non-specific encounter complexes with RNCs (regardless of nascent chain length), which then proceed through binding intermediates, thus allowing for discrimination of RNCs displaying a signal sequence. We have adapted our discussion by including the step of SRP discriminating RNCs via intermediates that we might not be detecting. Such intermediates might be what other groups have observed arriving and forming on RNCs with arrival rates that are insensitive to nascent chain length.

*The fact that the smFRET experiments were performed at “physiologically relevant SRP concentrations” is stated in many places throughout the manuscript. This overemphasis on the use of physiological SRP concentrations seems unnecessary and gives the misleading impression that smFRET experiments cannot be informatively performed at SRP concentrations greater than or lesser than physiological. Arrival and residence times, as well as changes in arrival and residence times, can be measured at any SRP concentration, physiological or otherwise, at which individual binding events can be detected. The SRP concentration at which the smFRET experiments are performed is relevant only in that, at higher SRP concentrations, arrival times are faster such that the number of individual binding events that are observed is larger. In other words, smFRET experiments performed at SRP concentrations greater than or lesser than physiological would be just as informative as those reported here at physiological SRP concentrations, so long as individual binding events can be observed at those SRP concentrations. In fact, smFRET experiments performed at SRP concentrations greater than physiological might even be desired in some instances (e.g., in experiments in which a functional signal sequence is not exposed) in that it would increase the probability that binding intermediates with average lifetimes on the order of or slightly shorter than the time resolution of the smFRET experiments could be observed. Thus, the authors should really tone down their overemphasis on the use of physiological SRP concentrations*.

We agree that our phrasing in the text was confusing. We did not mean to imply that studying association constants at different concentrations is not useful. We do believe that in this specific context, where the association rates proved to be variable and slower than expected, it was necessary to have SRP concentrations high enough so that the interactions could be detected, but not so high that they could be dismissed as maybe not happening *in vivo*. Additionally, the ability to reach such a physiologically relevant concentration is an important benefit of the method we present because other single molecule approaches, such as TIRFM, cannot achieve such a concentration and would have missed the binding events. We have therefore kept the emphasis on the benefits of studying this particular process at physiologically relevant concentrations.

*2) The second issue relates to how stalled versus translating RNCs were experimentally compared. Quoting from one of the referees: “The authors attribute the differences between their results and those with stalled RNCs to the fact the latter are not translating. There may be other reasons in addition to or instead of this. For example, stalling translation with a hydrophobic TMD exposed to bulk aqueous solvent in a PURE translation system may well result in aggregation, sticking to whatever is available, and other non-physiologic consequences. This seems far more likely to cause the observed heterogeneity and artifacts than the fact that ribosomes are stalled. I therefore think the comparison between the translating ribosomes with smFRET and stalled RNCs prepared by different methods and deposited into the ZMWs afterwards is flawed. The appropriate comparison would be to use truncated mRNA for 55-mer that is translated in the ZMW (exactly as the real-time experiment), then add the SRP after the translation. This would more directly compare stalled versus translating.” Given that a major conclusion put forth relates to this comparison, the proposed experiment would seem worthwhile and well within the reagents and tools available to the authors*.

To address this legitimate concern we performed the suggested experiment, which is a better way to compare stalled RNCs with actively translating RNCs. The results have replaced the previous Figure 1—figure supplement 3. The results obtained were almost identical to those we had previously included in the manuscript (in which the ribosomes translated up to the truncated 3’-end of the mRNA using the PURE translation system).

The data provide further evidence that the observed differences between stalled and actively translating RNCs are not likely to be due to PURE system specifics such as those raised by the reviewer, but to the fact that one set of RNCs is translating while the other is not, and as such might explore a range of additional non-functional conformational states.